# Unlocking the Uterine Code: Microbiota, Immune Cells, and Therapy for Recurrent Reproductive Failure

**DOI:** 10.3390/microorganisms12030547

**Published:** 2024-03-09

**Authors:** Svetla Blazheva, Svetlana Pachkova, Tatyana Bodurska, Petar Ivanov, Alexander Blazhev, Tzvetan Lukanov, Emiliana Konova

**Affiliations:** 1Department of Clinical Laboratory, Allergology and Clinical Immunology, Medical University—Pleven, 5800 Pleven, Bulgaria; svetla.blajeva@mu-pleven.bg (S.B.); tcvetan.lukanov@mu-pleven.bg (T.L.);; 2Medical Center—Clinical Institute for Reproductive Medicine, 5804 Pleven, Bulgaria810285@mu-pleven.bg (T.B.);; 3Department of Biology, Medical University—Pleven, 1 Kliment Ohridski Str., 5800 Pleven, Bulgaria

**Keywords:** endometrial immune cells, endometrial microbiota, recurrent implantation failure, recurrent pregnancy loss

## Abstract

The uterine microbiota has been the subject of increasing study, but its interaction with the local immune system remains unclear. Successful embryo implantation relies on endometrial receptivity, which is pivotal for immunological tolerance to fetal antigens and precise regulation of inflammatory mediators. Emerging data suggest a dynamic interplay between endometrial microflora and the immune system, making dysbiosis a potential determinant of pregnancy outcomes. Imbalances in the regulation of immune cells in the endometrium and decidua have been associated with infertility, miscarriage, and obstetric complications. A thorough comprehension of the immune system in the female reproductive tract shows potential for improving women’s health and pregnancy outcomes. The objective of this study was to evaluate the patterns of endometrial microbiota in patients with recurrent implantation failure (RIF) and recurrent pregnancy loss (RPL) and to explore their implications for endometrial immune cells and chronic endometritis (CE). Immune cells in biopsies from 107 RIF and 93 RPL patients were examined using flow cytometry. The endometrial microbial composition was analyzed using real-time polymerase chain reaction (RT-PCR). The research uncovered disrupted endometrial microbiota in most women with RIF and RPL, which was often associated with significant effects on lymphocytes, T cells, and uNK cells.

## 1. Introduction

The group of microorganisms that live in association with the human body is known as the human microbiota. This community includes eukaryotes, archaea, bacteria, and viruses, with bacteria being the most abundant and better-studied members. Conversely, the microbiome refers to the genes and genomes of this microbiota and their products within the host environment, thus referring to the entire habitat, including biotic and abiotic factors [1]. Described as the “ecological community of commensal, symbiotic, and pathogenic microorganisms that literally share our body space”, the human microbiome encompasses a diverse array of microorganisms [2]. Before the second half of the 20th century, the uterine cavity was considered sterile. However, advancements in next-generation sequencing of the 16S ribosomal RNA (rRNA) gene have unveiled the existence of an endometrial microbiota, predominantly featuring lactobacilli and other bacterial species [3,4]. It has been suggested that colonization of the uterus may occur by blood circulation from the oral cavity and intestine [5]. Additionally, there is a potential avenue for colonization through microorganisms attaching to human spermatozoa [6] and via assisted reproductive technology (ART) procedures [7].

The composition of the endometrial microbiota is the subject of active research, with some studies supporting the hypothesis of *Lactobacillus* spp. dominance [8,9,10,11,12,13,14] and others emphasizing the prevalence of different microorganisms. Some studies suggest that the endometrial microbiota may be dominated by *Acinetobacter* spp., *Pseudomonas* spp., and other microorganisms rather than *Lactobacillus* spp. [15,16,17,18,19,20]. These observations contrast with other studies confirming the high presence of *Lactobacillus* spp. in the endometrial microbiota, particularly in healthy women. Although the data presented remain heterogeneous and inconclusive, they highlight the need for further research to elucidate the complex composition and functions of the endometrial microbiota in different physiological states and pathologies.

Alterations in bacterial communities within the uterine cavity have been linked to reproductive complications, encompassing conditions such as infertility [21], endometriosis [22], chronic endometritis [23], and RIF [24,25]. Various hypotheses have emerged to elucidate the virulence mechanisms stemming from a dysbiotic endometrial microbiome. These mechanisms could be associated with the bacterial communities themselves, involving phenomena like mucin degradation, biofilm formation, and alterations in the pH of the female genital tract [26,27]. These mechanisms might be attributed to the host’s proinflammatory immune responses triggered by dysbiotic bacterial states. Such immune responses likely play pivotal roles in driving infertility [28]. Dysbiotic states in the vaginal microbiota can affect fertility through complex molecular interactions. For instance, Group B Streptococcus (GBS) produces β-hemolysin/cytolysin, which triggers inflammation and disrupts maternal–fetal barriers via Toll-like receptor (TLR) activation [29]. Bacteria associated with bacterial vaginosis (BV) stimulate proinflammatory responses through short-chain fatty acids (SCFAs), possibly via G protein-coupled receptor (GPCR) activation. Preterm birth is associated with increased proinflammatory responses caused by non-lactobacilli taxa in the vaginal microbiota [30]. Cytokine release is induced by lipopolysaccharide (LPS) from Gram-negative bacteria activating TLR4 [31,32]. Dysbiosis is correlated with elevated levels of IL-1β, IL-6, IL-8, and GM-CSF, which contribute to uterine inflammation and cervical remodeling [33,34,35,36].

The endometrial immune system plays a crucial role in both maintaining a healthy uterine environment for embryo implantation and preventing immune rejection of the semi-allogeneic fetus. This delicate balance is achieved through the dynamic equilibrium of the uterine microecology, which depends on the interaction between the endometrial microbiota, the immune system, and the endometrium itself [37].

Recent findings propose a correlation between early dysregulation of endometrial immune responses and both RIF and RPL [38]. During the implantation window, crucial uterine immune responses take place that allow the embryo to be accepted and prevent rejection, facilitating implantation, growth, and nourishment [39,40]. Immune cells of innate immunity, namely macrophages, dendritic cells, uterine natural killer cells (uNK cells), and innate lymphoid cells-1 and -3, infiltrate the endometrium [41]. Distinguishing uNK cells from their peripheral counterparts are differences in phenotype, cytokine profiles, limited cytotoxic potential, and the repertoire of activating and inhibiting receptors [42]. Regulatory T cells (Tregs) hold a dominant role in preserving immunological self-tolerance. They achieve this by regulating the adaptive system and preventing immune and autoimmune responses directed at self- and paternal alloantigens [43]. Aberrant mobilization or expression of immune cells has been observed through flow cytometry and immunohistochemistry in patients with histories of RIF and RPL [42,43,44]. This suggests that localized immune imbalances within the endometrium contribute to the failures in implantation and pregnancy [45].

Chronic endometritis (CE) denotes an inflammatory condition of the endometrial lining, often asymptomatic, characterized by elevated stromal cell density, surface-level endometrial edema, and the infiltration of endometrial stromal plasma cells (ESPCs) [46,47]. Ultrasound examinations are not sufficient for CE identification, but fluid hysteroscopy unveils specific endometrial alterations linked to CE, including stromal edema, focal hyperemia, focal or diffuse micro polyps, and endometrial hemorrhagic spots [48]. Currently, the gold standard technique for CE diagnosis involves endometrial biopsy coupled with immunohistochemical staining for CD138 (Syndecan-1, a cell surface proteoglycan) to detect plasma cells [49].

The primary objective of this study was to analyze the endometrial microbiota profiles in Bulgarian patients with recurrent implantation failure (RIF) and recurrent pregnancy loss (RPL) and to evaluate their influence on endometrial immune cells and endometrial receptivity. In addition, these patients were followed prospectively to assess the efficacy of endometrial analysis and subsequent treatment in improving the success rate of their first embryo transfer.

## 2. Materials and Methods

### 2.1. Study Design and Participant Recruitment

This prospective cohort study was conducted at the “Clinical Institute for Reproductive Medicine” in Pleven, Bulgaria. The research encompassed 107 patients with RIF and 93 patients with RPL who sought fertility treatment at our center for assisted reproductive technology (ART) between January 2021 and November 2023. The average age of the participants was 35.26 ± 4.54 years, ranging from 26 to 41 years.

Inclusion criteria:

1. The participants required the use of the intracytoplasmic sperm injection (ICSI) technique to create embryos;

2. The women had undergone at least three fresh or frozen cycles with the transfer of a minimum of four good-quality embryos but failed to achieve pregnancy or maintain two or more clinically recognized pregnancies beyond 22 weeks of gestation;

3. Both partners exhibited normal chromosome karyotypes, as determined via chromosome analysis of peripheral lymphocyte cultures. Conventional GTG banding was performed on phytohemagglutinin-stimulated peripheral blood lymphocytes, with a comprehensive analysis of 30–62 metaphases per individual (resolution 400–550 bands/haploid set), including karyotyping of at least 20 metaphases;

4. The ovarian reserve function was normal, indicated by follicle-stimulating hormone (FSH) levels below 12.5 mIU/mL and anti-Müllerian hormone (AMH) levels above 0.7 ng/mL;

5. The shape of the uterine cavity was verified as normal through ultrasound and/or hysteroscopy examinations;

6. No antibiotic treatment was administered within the month preceding specimen collection;

7. To prevent the introduction of dysbiotic microorganisms from the vagina and cervical canal into the uterus during biopsy, microbiological examination of a vaginal swab and RT-PCR Femoflor screen of cervical swab was performed;

8. A further inclusion criterion was a history of a regular menstrual cycle with a duration of 26 to 29 days.

Figure 1 provides a visual overview of the study design.

In the study cohort, participants self-reported a menstrual cycle of 26 to 29 days, indicating a regular menstrual cycle. Ovulation status was confirmed on the day of the study through an assessment of follicular development and endometrial thickness via transvaginal ultrasound prior to the performance of endometrial biopsy procedures.

Exclusion criteria involved factors known to contribute to implantation failure and pregnancy loss, untreated hydrosalpinx, and uterine malformations.

### 2.2. Ethical Approval

This study received ethical approval from decision of the Ethics Committee for Scientific Research at the Medical University of Pleven #698-KENID/03.06.2022. All participants in this study were provided with comprehensive information about this study and its procedures, and they willingly signed an informed consent form. All measurements and procedures were conducted in accordance with pertinent guidelines and regulations. It is important to note that this study did not involve any research products.

### 2.3. Sample Collection Procedure

Endometrial specimens were procured during the mid-luteal phase (days 21–22) of natural menstrual cycles, ensuring a gap of at least three months after the most recent unsuccessful ART cycle. The mid-luteal phase, which occurs approximately 7–9 days post-ovulation, is chosen for endometrial sampling because of its physiological significance. This phase marks the ‘implantation window’, characterized by optimal conditions for embryo attachment and implantation. High progesterone levels during this phase drive endometrial transformation, facilitating glandular secretion, stromal decidualization, and vascularisation, which are essential for the formation of a receptive endometrium. Sampling during this critical period provides valuable insight into the endometrial environment, which is critical for successful pregnancy establishment. Abnormalities in endometrial receptivity are associated with recurrent implantation failure (RIF) and pregnancy loss (RPL). Focusing on the mid-luteal phase improves study standardization and reproducibility, reduces variability across menstrual cycles, and ensures consistent sample collection procedures.

Following the placement of a sterile speculum, thorough cleaning of the vagina and cervix was conducted using 0.9% NaCl solution. The cervical mucus was carefully eliminated, and no antiseptic treatment was applied to the speculum or vaginal area. The patient abstained from any hygienic procedures, vaginal douches, topical medications, or probiotic preparations.

For specimen collection, a flexible, sterile catheter was introduced into the uterine cavity, utilizing two types of catheters: Malleable Stylet and Intra-uterine Insemination Catheter (Wallace^®^, Cooper Surgical, Inc., Shelton, CT, USA). No flushing was performed into the uterine cavity during sample collection to promote mucosal retrieval. Negative pressure was established by connecting a 1 mL syringe to the opposite end of the catheter, coupled with gentle catheter movements within the uterine cavity to aspirate uterine mucosa. Aspiration ceased upon the catheter’s return to the level of the internal os of the cervical canal. To prevent any potential contamination with vaginal fluid, the catheter was meticulously prevented from making contact with the vaginal walls.

Transcervical access for assessment of the endometrial microbiome may influence microbiological results. However, this is the only clinical method for endometrial assessment. A spectrum of microbiota exists in the female reproductive system, so even with contamination, microbial profiles remain consistent with the microbial environment of the uterine cavity. To ensure accurate result interpretation, it was imperative that the sample contained a substantial quantity of endometrial mucosa. Evaluation of the sample quantity was subjectively conducted by the performing physician. If a biopsy was deemed inadequate, a repeat biopsy was performed. The collected samples were processed within one hour of collection to maintain their integrity.

### 2.4. Sample Storage Protocol

The aspirated endometrial mucosa was preserved in a solution of 0.9% NaCl and promptly transported to the laboratory post-collection. Upon arrival, the sample underwent division into two equal portions for distinct processing.

The first portion was homogenized within a sterile 35 mm Medicon (BD Biosciences, Heidelberg, Germany) using a DAKO Medimachine tissue homogenizer (Becton Dickenson, Heidelberg, Germany). The homogenate was then aspirated into a sterile syringe and stored within saline solution. This portion was maintained within a temperature range of 2 °C to 8 °C for a maximum of 24 h before undergoing analysis through RT-PCR. In instances where analysis could not be conducted within 24 h, the material was subjected to freezing, capable of being stored at temperatures ranging from −18 °C to −22 °C for up to 1 month.

The second portion underwent homogenization within 1 mL of phosphate-buffered saline (PBS) within a 35 µm Medicon for a duration of 60 s. Subsequently, the obtained homogenate was subjected to filtration through a 70 µm syringe filcon (BD Biosciences, Germany). This filtrate was promptly prepared for analysis through flow cytometry.

### 2.5. DNA Extraction

DNA extraction was carried out using the PREP-NA-PLUS reagent kit (DNA Technology LLC; Moscow, Russia), following the prescribed protocol provided by the manufacturer.

### 2.6. Sample Analysis

Analysis of the endometrial microbiome samples was performed using the Femoflor^®^ 16 REAL-TIME PCR Detection Kit, developed by “DNA-Technology Research & Production” in the Moscow Region, Russia. This kit employs the polymerase chain reaction (PCR) nucleic acid amplification technique for the identification of lactobacilli, aerobic and anaerobic microorganisms, yeast-like fungi, and urogenital mycoplasms [50] with sensitivity of 99% and specificity of 93%.

To enhance the precision and specificity of the amplification reaction, a hot-start method is employed. This involves the preparation of the reaction mixture with two distinct layers, separated by a paraffin layer, or the use of Taq-polymerase that is blocked by antibodies. The Femoflor^®^ 16 kit (DNA Technology LLC; Moscow, Russia) capitalizes on a fluorescence-based modification of the PCR method. The PCR mix includes two target-specific probes that bear reporter fluorescent dyes (Fam and Hex) alongside quencher molecules. Once these probes hybridize with the target sequence, they become activated, resulting in an increase in fluorescence proportionate to the amplification of the target sequence. The intensity of this fluorescence is measured during each cycle of the reaction using an RT-PCR thermal cycler data collection unit and subsequently analyzed using RealTime_PCR v7.9 (DNA Technology LLC; Moscow, Russia).

The cumulative birth rate was calculated as the percentage of cases with a live birth relative to the total number of cases.

### 2.7. Analysis of Endometrial Immune Cells

Flow cytometry was used to examine lymphocyte subpopulations in endometrial biopsies. The analysis of leukocytes was conducted using a FACS Calibur flow cytometer (Becton Dickinson, Heidelberg, Germany) along with BD CellQuest™ Pro Software v 6.0 (Becton Dickinson). A set of specific monoclonal antibodies was utilized, including CD45 Per CP, CD34 FITC, CD16 PE, CD56 PE, and CD138 FITC (Exbio, Prague, Czech Republic).

The analysis involved three distinct tubes, each containing a different combination of antibodies:Tube 1: CD3 FITC/CD16+56 PE/CD45 Per CP

This tube enabled the evaluation of uterine/decidual-type uterine natural killer cells (uNK, CD3-, CD16-, CD56bright) and T cells (CD3+, CD16-, CD56-).
Tube 2: CD34 FITC/CD56 PE/CD45 Per CP

This tube allowed for the evaluation of progenitor natural killer cells (CD34+, CD16-, CD56+).
Tube 3: CD138 FITC/CD45 Per CP

This tube facilitated the evaluation of plasma cells (CD45+CD138+).

The resulting data were expressed as a percentage of CD45+ cells. More specifically, the analysis encompassed the following:-Leucocytes (CD45+), presented as a percentage relative to all endometrial cells (leucocytes, stromal cells, epithelial cells). By targeting CD45, we aimed to identify and quantify the total leukocyte population within the endometrium, providing an overview of the overall immune cell infiltration and composition;-Lymphocytes, macrophages, and neutrophils (CD45+, CD16+), presented as a percentage by gate and relative to all leukocytes;-Uterine/decidual-type uterine natural killer cells (uNK, CD3-, CD16-, CD56 bright), progenitor natural killer cells (CD34+ uNK cells, CD34+, CD16-, CD56+), and T cells (CD3+, CD16-, CD56-), presented as a percentage relative to all lymphocytes. CD56 and CD16 are markers commonly found on natural killer (NK) cells. NK cells play a crucial role in immune surveillance and regulation at the maternal–fetal interface. By targeting CD3, we aimed to quantify the population of T cells infiltrating the endometrium, as aberrations in T-cell subsets have been implicated in the pathogenesis of RIF and RPL;-Plasma cells (CD45+, CD138+), expressed as a percentage of total leukocytes and relative to total cells (leukocytes and stromal cells). CD138, also known as syndecan-1, is a cell marker specific for plasma cells, which are often associated with chronic inflammatory processes.

### 2.8. Statistical Analysis

The statistical analysis was performed utilizing IBM SPSS, version 26. To initiate the analysis, all parameters underwent assessment for the normality of distribution using the Shapiro–Wilk W test. For parametric data, the ANOVA test was employed, while nonparametric data were subjected to Spearman’s test. In cases where the test’s significance level was below 0.05 (*p* < 0.05), the null hypothesis of a normal distribution was rejected, signifying a divergence from normal data distribution. Comparisons between categorical variables were assessed with chi-square test. One-way ANOVA test was used for detailed analysis of immune cells between the groups. For statistical analyses, Prism software version 8.4.3 (GraphPad, GraphPad Software, San Diego, CA, USA) was utilized. Unpaired nonparametric comparisons were conducted using the Mann–Whitney U test. Additionally, the nonparametric Pearson correlation test was employed to assess correlations between two parameters.

## 3. Results

### 3.1. Pre-Treatment Pathogen Analysis Results

Vaginal and cervical swab tests were performed before endometrial biopsy. Pathogens associated with sexually transmitted infections, including *T. vaginalis*, *C. trachomatis*, *M. genitalium*, and *N. gonorrhoeae*, were tested in the vaginal swab, and bacteria associated with BV, herpes viruses, and *Ureaplasma* spp. in cervical secretions. Treatment was administered to 72 women based on the detection of various pathogens in these secretions, including those responsible for bacterial vaginosis, candidiasis, and aerobic vaginitis. The endometrial biopsy was performed after a control examination of vaginal and cervical secretions, which showed the absence of pathogenic microorganisms, and at least one month after the end of antibiotic therapy.

### 3.2. Group Classification Based on Molecular Genetic Testing

Based on the outcomes obtained from molecular genetic testing of the endometrial biopsies, a total of 200 women were classified into four distinct groups as follows:Group 1 includes women (*n* = 41) with a normal endometrial microbiota characterized by more than 90% *Lactobacillus* spp. presence and the absence of dysbiotic bacteria;Group 2 includes women (*n* = 57) whose biopsies exhibited a low bacterial mass, with the absence of both *Lactobacillus* spp. and dysbiotic bacteria;Group 3 includes women (*n* = 25) with a disturbed endometrial microbiota, marked by less than 90% *Lactobacillus* spp. presence and more than 10% dysbiotic bacteria.Group 4 includes women (*n* = 77) with severely disturbed microbiota, characterized by the absence of *Lactobacillus* spp. and more than 10% dysbiotic bacteria.

Patients with endometrial microbiota from Group 3 and Group 4 were treated with antibiotics according to the type of isolated microorganisms and antifungals. Topical and oral probiotics were also used in Group 2, Group 3, and Group 4.

At least 4 weeks after the last antibiotic, antifungal, or/and probiotic intake, a control endometrial sample was collected and tested by Femoflor test. Only those women whose PCR tests were negative for dysbiotic microorganisms were allowed to proceed with the next IVF procedure.

### 3.3. Results of Endometrial Microbiota Examination

The median with minimum and maximum age within groups categorized by endometrial microbiota type is depicted in Figure 2. Group 1 exhibited a mean age of 34.17 ± 4.45 years, Group 2 had a mean age of 35.32 ± 4.79 years, Group 3 showed a mean age of 35.64 ± 4.90 years, and Group 4 had a mean age of 35.69 ± 4.25 years.

The molecular genetic analysis of 200 endometrial biopsies in women experiencing RIF and RPL revealed a prevalence of severely impaired microbiota, characterized by the absence of *Lactobacillus* spp. and the presence of more than 10% dysbiotic bacteria.

Specifically, this pattern was observed in 77 cases, constituting 38.5% of the studied cohort. The second most frequent group (*n* = 57, 28.5%) lacked both dysbiotic microorganisms and lactobacilli. Conversely, 41 biopsies (20.5%) showed lactobacilli constituting over 90% of the microbiota. The lowest number of biopsies was in Group 2 (*n* = 25, 12.5%).

The prevalence of dysbiotic bacteria, identified either independently or in combination, in Group 3 and Group 4 was as follows: anaerobic and *Candida* spp. manifested with the highest frequency in 33 women, constituting 16.5% of the total cases (*n* = 200), as detailed in Table 1. There was no statistically significant difference in the distribution of the type of microorganisms in the studied endometrial microbiota between group 3 and group 4 (χ^2^(6, *n* = 102) = 8.757, *p* = 0.188). Additionally, aerobic, anaerobic microorganisms and *Candida* spp. were detected in 26 cases (13.0%). Specifically, anaerobic microorganisms were identified in 9% of biopsies (*n* = 18), followed by *Candida* spp. in 15 cases (7.5%). In the remaining instances, with an incidence of less than 7.5%, biopsies featuring a combination of aerobic microorganisms and *Candida* spp. were observed at 6%, while those with aerobic alone were at 3%, and a combination of aerobic and anaerobic microorganisms at 2%.

Table 2 shows the composition of the uterine microbiome in two groups of women: those with RIF and those with RPL. It further categorizes women within each group based on the state of their endometrial microbiome: normal EM, low biomass, disturbed EM, and severely disturbed EM. The percentage of women with impaired and severely impaired endometrial microbiota was similar in women with RIF (52.34%) and RPL (49.46%). Women with RIF were found to have a higher number of cases of *Candida* spp. (9.35%) than women with RPL (5.38%).

### 3.4. Results of Endometrial Immune Cells

The mean percentages and standard deviations of endometrial cells in the four groups are presented in Table 3.

A statistical analysis of the leukocyte and lymphocyte populations revealed significant differences in the percentages of lymphocytes (H (3) = 8.949, *p* = 0.030), uNK cells (H (3) = 13.846, *p* = 0.003), and T cells (H (3) = 12.860, *p* = 0.005) among the four study groups (Figure 3).

The analysis showed that T cells were statistically significantly lower in group 3 compared with group 4 (*p* = 0.026) and group 2 (*p* = 0.006).

The uNK cells were statistically significantly lower in Group 3 compared with Group 4 (*p* < 0.001), Group 2 (*p* < 0.003), and Group 1 (*p* < 0.002).

The T cells were statistically higher in Group 2 compared with Group 1 (*p* = 0.032), Group 3 (*p* < 0.001), and Group 4 (*p* < 0.006).

There were no significant differences in mean leukocyte percentages in our four groups (*p* > 0.05).

In group 3, 11 out of 25 biopsies were found to have a higher neutrophil percentage than the group mean, but only 1 had elevated plasma cells (7.29% relative to all cells). In Group 4, 30 out of 77 biopsies were found to have a higher neutrophil percentage than the group mean, but only 2 had elevated plasma cells (4.76% and 3.23% relative to all cells).

### 3.5. Results of Follow-Up of the Treatment Effect

After treatment of endometrial dysbiosis, patients were followed up. Table 4 shows cumulative birth rates after subsequent ART (ICSI) in the different groups. The highest cumulative birth rate was observed in a group of women where the endometrial microbiota was dominated by lactobacilli.

Of those who underwent treatment (*n* = 163), 56 had successful pregnancies and delivery of healthy newborns, and 24 had an ongoing pregnancy, meaning that the common cumulative post-treatment pregnancy rate of the followed patients was 56 out of 200 (28.0%).

Twenty-nine of the seventy-seven women in Group 4 had clinical pregnancies, and eleven of them are still in ongoing pregnancy, while 18 have delivered healthy newborns. Forty-one of them tested negative for hCG tests, and seven had miscarriages after the next ART (ICSI). Nine of the twenty-five women in Group 3 had clinical pregnancies, and two of these are still in ongoing pregnancy, while seven have delivered healthy newborns. Thirteen of them tested negative for hCG tests, and three had miscarriages after the next ART (ICSI). In Group 2, probiotic therapy resulted in pregnancy for 28 women, of whom 12 achieved successful pregnancies, 8 are currently experiencing ongoing pregnancies, and 8 had miscarriages.

*Candida* spp. was detected in four biopsies from Group 1. In these cases, antifungal treatment was administered. Treatment with antibiotics and probiotics, based on the identified causative agents in Group 3 and Group 4 (*n* = 102), resulted in the disappearance of dysbiotic pathogens in 41 women (40.2%) after initial therapy. After therapy, the microbiota was found to be normal in 25 women, with lactobacilli comprising over 90% and dysbiotic bacteria being absent, while in 16 women, the microbiota had low biomass. In these cases, probiotic therapy was prescribed.

## 4. Discussion

The aim of this study was to investigate the endometrial microbiota of Bulgarian patients with RIF and RPL and to assess its impact on endometrial immune cells. In addition, we planned to evaluate the potential benefits of endometrial analysis and treatment on the success of their first subsequent embryo transfer.

The complex and dynamic interactions between the endometrial microflora, the immune system, and the endometrium form a delicate equilibrium in the uterine microenvironment. Changes in any of these components can initiate a chain of events that disrupt this equilibrium and lead to a number of pathological changes in the endometrium [51]. The studies presented here reflect the current state of this microenvironment during the mid-luteal phase, representing the specific area from which the endometrial biopsy was obtained.

One potential mechanism is through the modulation of the local immune response. Dysbiotic changes in the endometrial microbiota may trigger an abnormal inflammatory response, leading to impaired implantation and an increased risk of pregnancy loss. This dysregulated immune response could result in altered cytokine profiles, compromised endometrial receptivity, and impaired embryo–maternal communication, all of which are critical for successful implantation and pregnancy maintenance [37].

Additionally, dysbiosis in the endometrial microbiota may directly impact the endometrial environment, affecting factors such as hormone levels, nutrient availability, and tissue remodeling processes [52]. These alterations can disrupt the delicate balance necessary for embryo implantation and placental development, contributing to RIF and RPL.

Furthermore, the endometrial microbiota may influence reproductive outcomes through interactions with the vaginal microbiota and systemic immune responses. Dysbiosis in the vaginal microbiota has been associated with adverse pregnancy outcomes, and emerging evidence suggests that crosstalk between the vaginal and endometrial microbiota may occur, affecting the local and systemic immune milieu in the reproductive tract [53].

The results of this study confirmed the key role of the normal endometrial microbiota, characterized by the presence of more than 90% *Lactobacillus* spp., to accomplish optimal implantation and embryo development [54].

A study by Moreno et al. found that lactobacilli predominated in endometrial fluid (>90%), followed by *Gardnerella*, *Streptococcus,* and *Bifidobacterium* [9]. *Lactobacillus* spp. (*L. crispatus*, *L. gasseri*, *L. iners*, *L. jensenii*) may play a protective role in maintaining the health of the vaginal and endometrial environment [26]. Lactobacilli maintain an acidic environment that has antibacterial, antiviral, and immunomodulatory properties. This, in turn, inhibits the invasion and colonization of pathogenic bacteria. [55]. Lactic acid produced by *Lactobacillus* spp. has been shown to elicit an anti-inflammatory response and inhibit the production of proinflammatory cytokines and chemokines triggered by Toll-like receptor (TLR) activation in cervical and vaginal epithelial cells under acidic conditions [56,57]. Furthermore, lactic acid can stimulate the secretion of the anti-inflammatory cytokine interleukin (IL)-10, diminish the production of the proinflammatory cytokine IL-12 in dendritic cells, and attenuate the cytotoxicity of natural killer cells [58]. The anti-inflammatory properties of lactic acid are also contingent upon the presence of organic acids synthesized by microorganisms to sustain vaginal health. This is primarily achieved through the upregulation of the anti-inflammatory cytokine IL-1RA, inhibition of the proinflammatory signaling of IL-1 cytokines, and a modest reduction in the production of proinflammatory cytokines such as IL-6 and macrophage inflammatory protein 3 alpha [57]. Hence, the interplay among flora, metabolites, and immunity within the healthy reproductive tract is crucial for preserving its overall health. Any imbalance among these components can disrupt the equilibrium of the reproductive tract.

Molecular genetic analysis of endometrial biopsies from women with RIF and RPL (*n* = 200) revealed that 77 biopsies had a severely impaired microbiota. This characteristic profile was characterized by a sharp absence of *Lactobacillus* spp. and a predominant presence of more than 90% dysbiotic bacteria. These findings highlight the potential impact of the microbiota on reproductive performance [3,4,59].

Cases within Group 3 demonstrate a comparatively lower frequency of confirmation. Further studies are needed to confirm this observed incidence and to investigate the causes. The endometrial microflora typically functions as a protective barrier, producing substances that hinder the attachment and proliferation of pathogenic microorganisms. However, in the presence of such pathogens, it can also initiate a protective immune response by producing inflammatory cytokines, chemokines, and antibacterial substances [51].

The absence of lactobacilli in Group 2 showed statistically significantly higher percentage values of T cells, confirming the involvement of lactobacilli in the regulation of T cells. A similar conclusion was reached in a study using flow cytometry analysis of uterine and decidual samples from mice. After treatment with lactobacilli, the samples showed a significant reduction in the percentage of CD8+ T cells [60]. On the other hand, this may be due to the interaction of NK crosstalk between decidual macrophages, NK, and T cells [61]. The interaction between NK cells and T cells plays an essential role in this context. NK cells are known for their ability to regulate the immune response, including the activity of T cells. In the absence of lactobacilli, which normally maintain the balance in the endometrial microflora, NK cells can influence the activity of T cells by secretion of galectin and glycodelin, increasing their percentage [62].

The leukocyte percentages we observed in our samples were consistent with those reported in previous studies, demonstrating the validity of our sampling and processing procedures. During the mid-luteal phase, leukocytes can account for up to 30% of the total stromal cell population [63].

The uNK cells are key decidual immune cells that play a crucial role in establishing pregnancy. During the preovulatory phase of the menstrual cycle, uNK cells are weakly present in the endometrium. As progesterone levels rise during the secretory phase, uNK cell numbers surge dramatically. If pregnancy is established, uNK cells further proliferate, constituting up to 60–90% of decidual immune cells [64,65]. Early in human pregnancy, uNK cells become prominent at the fetomaternal interface, gradually diminishing during the middle and late stages. This decline is indicative of uNK cells fulfilling essential immune regulatory functions early in pregnancy. Our study’s results revealed a notably lower percentage of uNK cells in Group 3, where the mean percentage of plasma cells was the highest compared to the other groups. This observation aligns with a study by Matteo and colleagues examining 23 endometrial biopsies from women with reproductive disorders [66]. In cases of chronic endometritis, a significant decrease in the percentage of uNK cells was reported. These observations highlight the relationship between the status of the endometrial microbiota, the presence of plasma cells, and the percentage of uNK cells, providing a broader understanding of the interrelationship between these factors in the context of reproductive health. Women with disrupted microbiota displayed lower levels of lymphocytes and uNK cells, key regulators of the immune system, and their depletion may contribute to the development of implantation failure and recurrent pregnancy loss [41].

Neutrophils play a protective role through degranulation, phagocytosis, or the formation of neutrophil extracellular traps. Their function is related to the production of defensins, small cationic proteins that provide antibacterial, antiviral, and antifungal protection. They can also change their phenotype upon microbiota loading. During infection, neutrophils rapidly infiltrate the endometrium, enhancing innate immune defenses [5]. While a statistically significant difference between the groups was not observed, Table 3 indicates that Group 3 and Group 4 displayed higher mean neutrophil percentages. These findings align with previous studies demonstrating a substantially lower percentage of neutrophils in the normal microbiome compared to the impaired [3,4].

The lack of a statistically significant difference between plasma cells in the four study groups was not expected. This guides us that it is necessary to include additional factors and components that may be involved in chronic endometritis (viruses, autoimmune, and anti-inflammatory). Furthermore, both innate and adaptive immune pathways may be independently involved in conditions of dysbiosis and acute or chronic endometritis.

Endometrial dysbiosis is a disorder of the normal bacterial diversity in the uterus. This disorder can lead to inflammation and damage to the uterine lining, which can make it difficult for the embryo to implant and develop. The study also found that endometrial dysbiosis is associated with changes in the lymphocyte populations.

Statistical analysis showed a significant decrease in lymphocyte levels in Group 3 compared to Group 4 (*p* = 0.026) and Group 2 (*p* = 0.006). This finding is significant in that it justifies our expectation that differences in endometrial microbiota type dictate changes in immune responses. Although the exact mechanisms underlying this difference require further investigation, it highlights the importance of considering lymphocyte dynamics in the context of the endometrial microenvironment. Future research efforts could go deeper into this area by shedding light on new aspects of this study.

The presented results of the therapeutic approach after assessment of the endometrial microbiota confirm the positive assessment of this type of diagnostic procedure for the clinical outcome of assisted reproductive technology (ART) procedures.

In our study, the therapeutic strategy was tailored to the microbial findings from the endometrial biopsies. Specifically, when aerobic bacteria were isolated, we used quinolone monotherapy. In cases where only anaerobic microorganisms were found, we used Flagyl or Clindamycin monotherapy. For simultaneous isolation of aerobic and anaerobic microorganisms, we used quinolones together with Flagyl. In addition, an antifungal agent was added to this combination when *Candida* spp. were isolated.

The rationale behind these specific treatments was to effectively target the identified microbial species, taking into account their aerobic or anaerobic nature. By tailoring the therapeutic approach to the microbial profile of each patient, we aimed to optimize treatment efficacy and minimize the risk of resistance development.

The overall cumulative birth rate after treatment was 28.0%, which is also seen in other trials [67]. The results for cumulative birth rate were similar in Groups 3 and 4 (Table 4). This is not surprising to us as the therapeutic approach in the two groups was similar. In Group 1, the cumulative incidence of pregnancy was the highest despite antifungal treatment in only four women. This observation can be explained by the proven positive effect of endometrial brushing, which has been well-studied [68]. The exact mechanism has been explored by Gnainsky et al. [69], who demonstrated that an endometrial biopsy enhances its receptivity by attracting inflammatory agents, such as interleukin-6 (IL-6), IL-8, IL-15, CXC-chemokine ligand 1, or osteopontin and tumor necrosis factor (TNF).

Probiotic therapy has emerged as an effective tool in a large number of cases where the lactobacilli are less than 90%. This effect points to the urgent need for the active development of probiotic formulations specifically adapted for the treatment of endometrial dysbiosis.

Therapy of women with antibiotics and probiotic agents, alone or in combination with antifungals, has contributed to positive outcomes after ART procedures and has been shown to play a role in the recovery of endometrial microbiome balance.

## 5. Limitations

This study compares endometrial immune cells in women experiencing reproductive issues with different microbiota types. It is important to note that a control group consisting of women with a normal microbiome and without reproductive problems was not included, which limits our ability to fully understand immune cell dynamics across diverse contexts. In addition, including a control group of healthy women of childbearing age without reproductive issues could provide valuable insights for selecting appropriate immunomodulatory therapy. Another limitation of our study is that it only focuses on the relative values of decidual immune cells without considering their functional characteristics and activation status. Furthermore, the lack of information on metabolites is another limitation, as these molecules are essential in evaluating the endometrial microenvironment.

## 6. Implication for Clinical Practice

Understanding the implications of our findings for clinical practice is paramount. Integrating the analysis of endometrial microbiota into routine clinical assessments can offer valuable insights into optimizing treatment strategies, particularly in assisted reproductive technology (ART) protocols. By leveraging microbiota profiles, clinicians may personalize therapeutic approaches, potentially improving treatment outcomes for patients experiencing reproductive challenges. However, the implementation of microbiota analysis in clinical practice may pose challenges, including the development of standardized guidelines and addressing logistical considerations. Nonetheless, by embracing these insights, healthcare providers can enhance their ability to tailor interventions and improve patient care in reproductive medicine.

## 7. Conclusions

The findings indicate a clear association between endometrial dysbiosis and notable alterations in endometrial immune cells, particularly impacting lymphocytes, T cells, and uNK cells. The administration of therapy involving antibiotics and probiotic agents, either independently or in conjunction with antifungals, has demonstrated favorable outcomes in assisted reproductive technology (ART) procedures. Moreover, this therapeutic approach has proven instrumental in restoring a balanced endometrial microbiota.

## Figures and Tables

**Figure 1 microorganisms-12-00547-f001:**
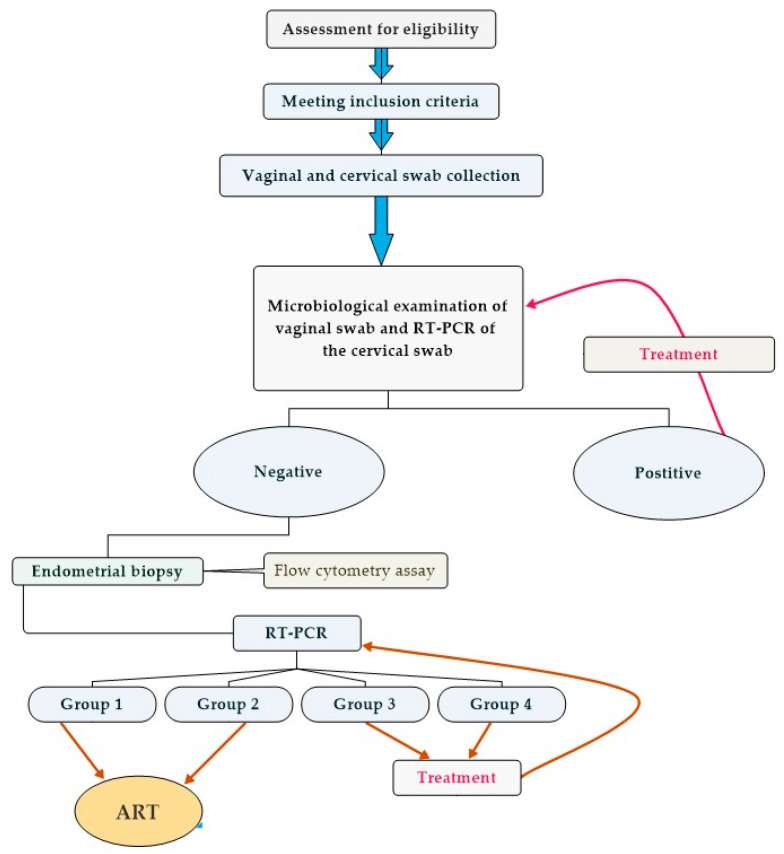
Diagram of this study and distribution of the population studied. The four groups in this study are women with normal endometrial microbiota (Group 1), low bacterial mass (Group 2), disturbed microbiota (Group 3), and severely disturbed microbiota (Group 4). ART—assisted reproductive technology.

**Figure 2 microorganisms-12-00547-f002:**
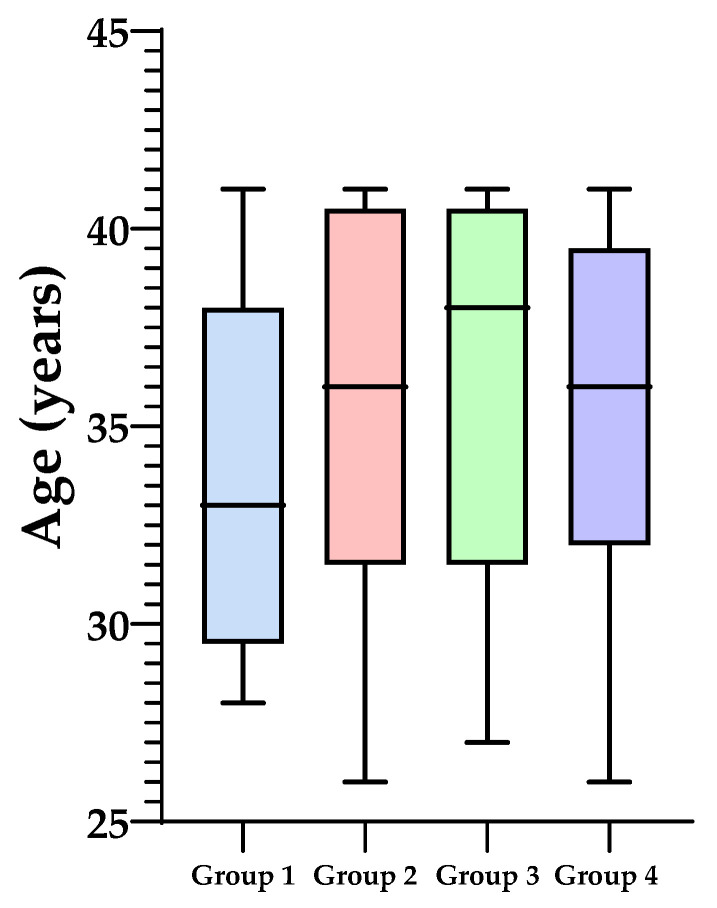
The median with minimum and maximum age within groups categorized by endometrial microbiota type.

**Figure 3 microorganisms-12-00547-f003:**
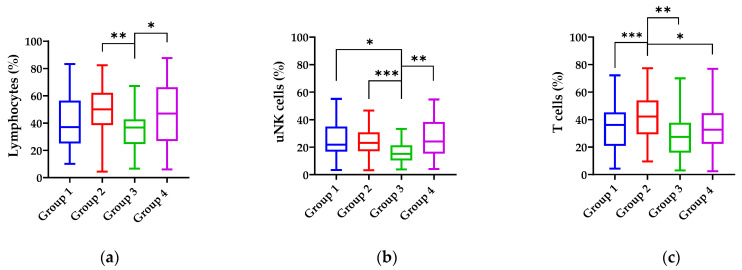
Comparative analysis of the percentage of lymphocytes (**a**), uNK cells (**b**), and T cells (**c**) in the cell suspensions from biopsies compared in different groups according to endometrial microbiota. Bars represent median and interquartile range. Statistically significant differences are presented as follows: (**a**) * *p* = 0.026; ** *p* = 0.006; (**b**) * *p* = 0.002; ** *p* < 0.001; *** *p* = 0.003; (**c**) * *p* = 0.006; ** *p* = 0.001, *** *p* = 0.032.

**Table 1 microorganisms-12-00547-t001:** Composition of the endometrial microbiota in 200 biopsies, categorized by groups of endometrial microbiota and the presence of different microorganisms.

	Endometrial Microbiota Type Number of Biopsies (%)				
Microbiota Profile	Normal EM	Low Biomass	Disturbed EM	Severely Disturbed EM	Total
	41 (20.5)	57 (28.5)	25 (12.5)	77 (38.5)	200 (100)
**Aerobic microorganisms**	-	-	3 (1.5)	3 (1.5)	6 (3)
**Anaerobic microorganisms**	-	-	3 (1.5)	15 (6.5)	18 (9.0)
**Aerobic and anaerobic** **microorganisms**	-	-	2 (1.0)	2 (1.0)	4 (2.0)
***Candida* spp.**	4 (2.0)	4 (2.0)	1 (0.5)	6 (3.0)	15 (7.5)
**Aerobic microorganisms and** ***Candida* spp.**	-	-	1 (0.5)	11 (5.5)	12 (6.0)
**Anaerobic and *Candida* spp.**	-	-	7 (3.5)	26 (13.0)	33 (16.5)
**Aerobic, anaerobic** **microorganisms, and *Candida* spp.**	-	-	10 (3.7)	16 (8.0)	26 (13.0)

**Table 2 microorganisms-12-00547-t002:** Uterine microbiome composition in groups with RIF and RPL.

		Aerobic Microorganisms (n)	Anaerobic Microorganisms (n)	Aerobic and anaerobic Microorganisms (n)	*Candida* spp. (n)	Aerobic Microorganisms and *Candida* spp. (n)	Anaerobic and *Candida* spp. (n)	Aerobic, Anaerobic Microorganisms, and *Candida* spp. (n)	Delivery (n)	Ongoing Pregnancy (n)
**RIF (*n* = 107)**	**Normal EM (*n* = 22)**	0	0	0	4	0	0	0	9	2
	**Low biomass (*n* = 29)**	0	0	0	4	0	0	0	4	3
	**Disturbed EM (*n* = 15)**	1	3	0	1	1	5	4	5	1
	**Severely disturbed EM (*n* = 41)**	2	7	2	1	5	17	7	9	7
	**Total RIF**	**3**	**10**	**2**	**10**	**6**	**22**	**11**	**27**	**13**
**RPL (*n* = 93)**	**Normal EM (*n* = 19)**	0	0	0	0	0	0	0	10	1
	**Low biomass (*n* = 28)**	0	0	0	4	0	0	0	8	5
	**Disturbed EM (*n* = 10)**	2	0	0	0	0	2	6	2	1
	**Severely disturbed EM (*n* = 36)**	1	8	2	1	6	9	9	9	4
	**Total RPL**	**3**	**8**	**2**	**5**	**6**	**11**	**15**	**29**	**11**

**Table 3 microorganisms-12-00547-t003:** Endometrial immune cells across different endometrial microbiota types in RIF and RPL groups. Mean and standard deviation analysis.

	RIF					RPL				
	Normal EM *n* = 22	Low Biomass *n* = 29	Disturbed EM *n* = 15	Severely Disturbed EM *n* = 41	Total RIF *n* = 107	Normal EM *n* = 19	Low Biomass *n* = 28	Disturbed EM *n* = 10	Severely Disturbed EM *n* = 36	Total RPL *n* = 93
**Age (years)**	35.86 ± 4.27	35.17 ± 4.88	37.07 ± 4.717	36.49 ± 3.99	36.08 ± 4.39	32.21 ± 3.9	35.46 ± 4.78	33.5 ± 4.58	34.78 ± 4.41	34.32 ± 4.54
**Leucocytes (%)**	17.71 ± 10.98	25.09 ± 15.73	23.12 ± 16.33	25.56 ± 15.29	18.26 ± 12.88	19.34 ± 14.99	22.92 ± 13.81	12.13 ± 9.72	16.62 ± 11.04	23.55 ± 14.85
**Lymphocytes (%)**	43.35 ± 19.64	50.77± 18.47	33.68 ± 11.42	44.61 ± 24.75	44.49 ± 21.04	38.94 ± 20.14	45.96 ± 18.77	38.46 ± 24.79	46.96 ± 21.76	44.11 ± 20.89
**Macrophages (%)**	31.60± 19.67	31.22 ± 18.86	36.32 ± 22.01	32.74 ± 21.96	32.6 ± 20.48	42.61 ± 27.17	26.72 ± 14.5	29.59 ± 24.63	27.89 ± 17.72	30.72 ± 20.55
**Neutrophils (%)**	21.14 ± 9.74	17.78 ± 2.45	29.75 ±15.98	22.02 ± 19.84	21.77 ± 17.72	17.30 ± 16.73	25.14 ± 18.69	31.12 ± 27.22	24.58 ± 19.72	23.97 ± 19.82
**T cells (%)**	34.28 ± 14.85	44.63 ± 15.2	26.86 ± 15.95	35.03 ± 18.09	36.33 ± 17.17	34.69 ± 18.54	40.27 ± 16.98	33.41 ± 20.39	34.18 ± 17.46	36.03 ± 17.79
**uNK cells (%)**	27.93 ± 12.44	23.93 ± 10.69	16.87 ± 9.81	26.78 ± 13.99	24.85 ± 12.65	23.08 ± 12.04	25.44 ± 13.74	17 ± 6.19	28.29 ± 15.98	25.15 ± 14.02
**CD34+ uNK cells (%)**	3.63 ± 4.14	2.41 ± 3.23	1.22 ± 2.18	2.1 ± 2.49	2.37 ± 3.11	2.44 ± 3.52	2.46 ± 3.51	2.07 ± 1.59	1.98 ± 2.35	2.23 ± 2.91
**Plasma cells * (%)**	2.70 ± 4.03	3.46 ± 6.48	6.31 ± 10.57	2.26 ± 5.38	3.24 ± 6.46	2.2 ± 2.44	1.18 ± 1.52	2.18 ± 5.39	2.12 ± 2.9	1.86 ± 2.84
**Plasma cells ** (%)**	0.64 ± 1.2	0.59 ± 1.19	0.92 ± 2.07	0.56 ± 1.93	0.64 ± 1.62	0.39 ± 0.51	0.29 ± 0.54	0.8 ± 2.04	0.53± 0.77	0.46 ± 0.89

* Plasma cells (CD45+, CD138+), presented as a percentage relative to all leukocytes and ** relative to all cells (leukocytes and stromal cells).

**Table 4 microorganisms-12-00547-t004:** Pregnancy outcomes following ART procedures by group.

Patients	Negative hCG Test	Pregnancy Loss	Ongoing Pregnancy	Delivery	Cumulative Birth Rate
**Normal EM** ***n* = ** **41**	18	1	3	19	46.34%
**Low biomass** ***n* = ** **57**	29	8	8	12	21.05%
**Disturbed EM** ***n* = ** **25**	13	3	2	7	28.0%
**Severely** **disturbed EM** ***n* = ** **77**	41	7	11	18	23.37%
**Total *n* = 200**	101	19	24	56	28.0%

## Data Availability

Data are contained within the article.

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
