# Peer review of "Unlocking the Uterine Code: Microbiota, Immune Cells, and Therapy for Recurrent Reproductive Failure"

_microorganisms, 2024, doi:10.3390/microorganisms12030547_

Round 1
Reviewer 1 Report
Comments and Suggestions for Authors
This manuscript investigates the impact of endometrial microbiota on reproductive health in Bulgarian patients experiencing recurrent implant failure (RIF) and recurrent pregnancy loss (RPL). Here's an overall comment on the manuscript:
Introduction The introduction sets a solid foundation, delineating the distinction between microbiota and microbiome and emphasizing the importance of Lactobacillus spp. in reproductive health. It effectively outlines the study's rationale, linking endometrial microbiota to reproductive outcomes. To enhance the quality of this manuscript, several critical comments and suggestions for improvement are offered:
A. More Recent References: Given the rapid advancements in microbiome research, incorporating more recent studies could update the narrative, especially those published after 2020. This would ensure the manuscript reflects the latest discoveries and theories in the field.
B. Detailed Mechanisms: The introduction mentions dysbiotic states triggering proinflammatory immune responses but could delve deeper into the specific mechanisms by which these states affect fertility. Providing examples of known pathways or molecular interactions would enhance the reader's understanding of the complex interplay between the microbiome and reproductive health.
C. Addressing Controversies and Debates: Highlighting ongoing debates or controversies within the field, such as differing opinions on the sterility of the uterus or the impact of ART on microbial colonization, would add depth to the discussion and acknowledge the complexity of the subject matter.
Materials and Methods This section is detailed, outlining the study's prospective cohort design, participant recruitment, ethical considerations, sample collection, and analysis procedures. The inclusion and exclusion criteria are clearly defined, ensuring a focused study population. However, elaborating on methodological choices, such as the specific timing for sample collection and statistical analysis justification, could enhance clarity and reproducibility. To improve the quality of the manuscript, consider the following critical comments:
A. Study Population Clarity: While the average age and the range of ages of the participants are provided, including the distribution of ages or median age might offer a clearer picture of the population studied. It's beneficial to understand the variability within your cohort.
B. Sample Collection Procedure: The detailed description of the sample collection procedure enhances the reproducibility of the study. However, discussing the choice of the mid-luteal phase for sample collection and its significance in the context of RIF and RPL would provide deeper insight into the timing of these procedures.
C. Flow Cytometry: The use of flow cytometry to examine lymphocyte subpopulations is a strong aspect of the methodology. Clarifying why specific monoclonal antibodies were chosen and how they contribute to understanding the endometrial immune environment in RIF and RPL would enhance this section.
Results The results section presents comprehensive findings on the endometrial microbiota composition and immune cell profiles in RIF and RPL patients, followed by the outcomes of therapeutic interventions. The statistical analysis of the microbiota and immune cells provides insights into the complex interplay between endometrial health and reproductive outcomes. To improve the quality and impact of this section, consider the following critical comments:
A. Linking Microbiota Findings to Clinical Outcomes: The results show a notable variance in microbiota composition across different groups. However, the manuscript could elaborate on how these variations correlate with the clinical outcomes observed after treatment. Discussing the potential mechanisms by which the endometrial microbiota influences RIF and RPL outcomes would provide a deeper understanding of the study's implications.
B. Treatment Effectiveness: The follow-up results after treatment interventions offer valuable insights into therapeutic strategies for RIF and RPL. However, elaborating on the rationale behind choosing specific treatments for each group and discussing the implications of these treatment outcomes in the broader context of managing RIF and RPL would strengthen the discussion.
C. Methodological Considerations for Future Studies: Acknowledging any limitations in the methodology used for microbiota and immune cell analysis, such as potential biases introduced by the transcervical sample collection method or the specificity and sensitivity of the Femoflor® 16 REAL-TIME PCR Detection Kit, would provide a more balanced view of the results.
Discussion The discussion effectively ties together the study's findings, situating them within the broader context of current research. It highlights the protective role of Lactobacillus spp., the impact of dysbiosis on immune response, and the potential of targeted therapies to improve ART outcomes. To further refine and enhance the quality of this discussion, consider the following points:
- Mechanisms Underpinning Observations: The discussion mentions the protective role of Lactobacillus spp. and the detrimental impact of a dysbiotic microbiota. Expanding on the biological mechanisms, possibly by referring to studies that have elucidated how Lactobacillus spp. Influencing immune regulation within the endometrium would add depth. For instance, discussing the specific immune pathways modulated by Lactobacillus spp. and how dysbiosis disrupts these pathways could provide insights into potential therapeutic targets.
- Therapeutic Interventions: The discussion benefits from detailing the outcomes of therapeutic interventions. Further elaboration on why certain therapies were chosen based on the microbiota profile and how these interventions align with current best practices in managing RIF and RPL could enhance understanding. Additionally, discussing the potential for personalized therapeutic approaches based on endometrial microbiota profiling could highlight the study's clinical implications.
- Addressing the Limitations: The limitations section is crucial for understanding the study's scope and the interpretation of its findings. Expanding on how these limitations might affect the study's conclusions and suggesting specific future research directions to address these gaps would be beneficial. For instance, discussing how the inclusion of a control group of women without reproductive issues could help delineate the normal variations in endometrial microbiota and immune cell dynamics across the menstrual cycle.
- Implications for Clinical Practice: While the therapeutic outcomes are promising, discussing the practical implications for routine clinical practice would be valuable. This could include considerations for integrating endometrial microbiota analysis in ART protocols, potential challenges in adopting these diagnostic and therapeutic approaches, and standardized guidelines based on microbiota profiles.
Author Response
Dear Reviewer,
Thank you very much for all your hard work and attention in reviewing our manuscript.
The authors would like to thank the Reviewer for all the remarks, comments, and recommendations that we have been able to take into consideration with the entire team of authors who have signed this work. We have tried, to the best of our ability, to apply and consider all the remarks and comments that you have sent to us. We hope that our responses will satisfy you.
We are sending you an improved manuscript, which has significantly improved.
Best regards:
Alexander Blazhev (Correspondent Author)
Introduction
Recommendation A. More Recent References: Given the rapid advancements in microbiome research, incorporating more recent studies could update the narrative, especially those published after 2020. This would ensure the manuscript reflects the latest discoveries and theories in the field.
Response to recommendation A. Thank you for your insightful recommendation. We recognize the importance of staying abreast of the latest advances in microbiome research to ensure the accuracy and currency of our manuscript. In response to your suggestion, we have conducted a thorough review and included studies published after 2020 that contribute to our understanding of the topic.
Recommendation B. Detailed Mechanisms: The introduction mentions dysbiotic states triggering proinflammatory immune responses but could delve deeper into the specific mechanisms by which these states affect fertility. Providing examples of known pathways or molecular interactions would enhance the reader's understanding of the complex interplay between the microbiome and reproductive health.
Response to recommendation B: Thank you for this recommendation. These information has been added to manuscript:
“Dysbiotic states in the vaginal microbiota can affect fertility through complex molecular interactions. For instance, Group B Streptococcus (GBS) produces β-hemolysin/cytolysin, which triggers inflammation and disrupts maternal-fetal barriers via toll-like receptor (TLR) activation [29]. Bacteria associated with bacterial vaginosis (BV) stimulate proinflammatory responses through short-chain fatty acids (SCFAs), possibly via G protein-coupled receptor (GPCR) activation. Preterm birth is associated with increased proinflammatory responses caused by non-lactobacilli taxa in the vaginal microbiota [30]. Cytokine release is induced by lipopolysaccharide (LPS) from Gram-negative bacteria activating TLR4 [31,32]. Dysbiosis is correlated with elevated levels of IL-1β, IL-6, IL-8, and GM-CSF, which contribute to uterine inflammation and cervical remodeling [33-36].“
Recommendation C.: Addressing Controversies and Debates: Highlighting ongoing debates or controversies within the field, such as differing opinions on the sterility of the uterus or the impact of ART on microbial colonization, would add depth to the discussion and acknowledge the complexity of the subject matter.
Response to recommendation C: Thank you for your recommendation. We appreciate the suggestion to address ongoing debates and controversies within the field, as it would indeed add depth to the discussion and acknowledge the complexity of the subject matter. We add a new paragraph addressing the controversy over the composition of the endometrial microbiota:
“The composition of the endometrial microbiota is the subject of active research, with some studies supporting the hypothesis of Lactobacillus spp. dominance [8-14] and others emphasising the prevalence of different microorganisms. Some studies suggest that the endometrial microbiota may be dominated by Acinetobacter spp., Pseudomonas spp. and other microorganisms rather than Lactobacillus spp. [15-20]. These observations contrast with other studies confirming the high presence of Lactobacillus spp. in the endometrial microbiota, particularly in healthy women. Although the data presented remain heteroge-neous and inconclusive, they highlight the need for further research to elucidate the com-plex composition and functions of the endometrial microbiota in different physiological states and pathologies.”
Materials and Methods
Thank you for all your important comments. According to your review and suggestion from the rest of the reviewers, section 2. Materials and methods will be expanded and improved.
Recommendation A. Study Population Clarity: While the average age and the range of ages of the participants are provided, including the distribution of ages or median age might offer a clearer picture of the population studied. It's beneficial to understand the variability within your cohort.
Response to recommendation A.
Thank you for this recommendation. An additional graph in 3. Results, where each bar represents the median and interquartile range, shows the age distribution in the groups.
Figure 1. The median with minimum and maximum age within groups categorised by endometrial microbiota type.
- Sample Collection Procedure: The detailed description of the sample collection procedure enhances the reproducibility of the study. However, discussing the choice of the mid-luteal phase for sample collection and its significance in the context of RIF and RPL would provide deeper insight into the timing of these procedures.
Response to B Thank you for this comment. We have included the additional clarification on the sampling procedure in the manuscript.
‘’The mid-luteal phase, which occurs approximately 7-9 days post-ovulation, is chosen for endometrial sampling because of its physiological significance. This phase marks the 'implantation window', characterised by optimal conditions for embryo attachment and implantation. Sampling during this critical period provides valuable insight into the endometrial environment, which is critical for successful pregnancy establishment. Focusing on the mid-luteal phase improves study standardisation and reproducibility, reduces variability across menstrual cycles and ensures consistent sample collection procedures.’’
- Flow Cytometry: The use of flow cytometry to examine lymphocyte subpopulations is a strong aspect of the methodology. Clarifying why specific monoclonal antibodies were chosen and how they contribute to understanding the endometrial immune environment in RIF and RPL would enhance this section.
Response to recommendation C. Thank you very much for your valuable comment. The choice of monoclonal antibodies for flow cytometry analysis was guided by their ability to target specific cell surface markers that are characteristic of distinct lymphocyte subpopulations relevant to endometrial immune profiling in the context of RIF and RPL. I present to you the changes in the text that complement and clarify the rationale for the choice of monoclonal antibodies for flow cytometric analysis.
„- Leucocytes (CD45+), presented as a percentage relative to all endometrial cells (leucocytes, stromal cells, epithelial cells). By targeting CD45, we aimed to identify and quantify the total leukocyte population within the endometrium, providing an overview of the overall immune cell infiltration and composition.
- Lymphocytes, macrophages, and neutrophils (CD45+, CD16+), presented as a percentage by gate and relative to all leukocytes.
- Uterine/decidual type uterine natural killer cells (uNK, CD3-, CD16-, CD56 bright), progenitor natural killer cells (CD34+ uNK cells, CD34+, CD16-, CD56+), and T cells (CD3+, CD16-, CD56-), presented as a percentage relative to all lymphocytes. CD56 and CD16 are markers commonly found on natural killer (NK) cells. NK cells play a crucial role in immune surveillance and regulation at the maternal-fetal interface. By targeting CD3, we aimed to quantify the population of T cells infiltrating the endometrium, as aberrations in T cell subsets have been implicated in the pathogenesis of RIF and RPL.
- Plasma cells (CD45+, CD138+), expressed as a percentage of total leukocytes and relative to total cells (leukocytes and stromal cells). CD138, also known as syndecan-1, is a cell marker specific for plasma cells, which are often associated with chronic inflammatory processes. “
Results The results section presents comprehensive findings on the endometrial microbiota composition and immune cell profiles in RIF and RPL patients, followed by the outcomes of therapeutic interventions. The statistical analysis of the microbiota and immune cells provides insights into the complex interplay between endometrial health and reproductive outcomes. To improve the quality and impact of this section, consider the following critical comments:
- Linking Microbiota Findings to Clinical Outcomes: The results show a notable variance in microbiota composition across different groups. However, the manuscript could elaborate on how these variations correlate with the clinical outcomes observed after treatment. Discussing the potential mechanisms by which the endometrial microbiota influences RIF and RPL outcomes would provide a deeper understanding of the study's implications.
Response to A: Thank you for the apparent oversight in the discussion of the data. We have added the necessary clarification in the 3. Results section, 3.4.Results of follow-up of the treatment effect:
‘’the highest cumulative birth rate was observed in a group of women (group 1) where the endometrial microbiota was dominated by lactobacilli’’.
and in 4. Discussion
‘’One potential mechanism is through the modulation of the local immune response. Dysbiotic changes in the endometrial microbiota may trigger an abnormal inflammatory response, leading to impaired implantation and increased risk of pregnancy loss. This dysregulated immune response could result in altered cytokine profiles, compromised endometrial receptivity, and impaired embryo-maternal communication, all of which are critical for successful implantation and pregnancy maintenance [1].
Additionally, dysbiosis in the endometrial microbiota may directly impact the endometrial environment, affecting factors such as hormone levels, nutrient availability, and tissue remodeling processes [2]. These alterations can disrupt the delicate balance necessary for embryo implantation and placental development, contributing to RIF and RPL.
Furthermore, the endometrial microbiota may influence reproductive outcomes through interactions with the vaginal microbiota and systemic immune responses. Dysbiosis in the vaginal microbiota has been associated with adverse pregnancy outcomes, and emerging evidence suggests that crosstalk between the vaginal and endometrial microbiota may occur, affecting the local and systemic immune milieu in the reproductive tract [3].’’
- Treatment Effectiveness: The follow-up results after treatment interventions offer valuable insights into therapeutic strategies for RIF and RPL. However, elaborating on the rationale behind choosing specific treatments for each group and discussing the implications of these treatment outcomes in the broader context of managing RIF and RPL would strengthen the discussion.
Response to recommendation B. Thank you very much for your valuable feedback. We added this information in the text:
‘’In our study, the therapeutic strategy was tailored to the microbial findings from the endometrial biopsies. Specifically, when aerobic bacteria were isolated, we used quinolone monotherapy. In cases where only anaerobic microorganisms were found, we used Flagyl or Clindamycin monotherapy. For simultaneous isolation of aerobic and anaerobic microorganisms, we used quinolones together with Flagyl. In addition, an antifungal agent was added to this combination when Candida spp. were isolated.
The rationale behind these specific treatments was to effectively target the identified microbial species, taking into account their aerobic or anaerobic nature. By tailoring the therapeutic approach to the microbial profile of each patient, we aimed to optimise treatment efficacy and minimise the risk of resistance development.’’
- Methodological Considerations for Future Studies: Acknowledging any limitations in the methodology used for microbiota and immune cell analysis, such as potential biases introduced by the transcervical sample collection method or the specificity and sensitivity of the Femoflor® 16 REAL-TIME PCR Detection Kit, would provide a more balanced view of the results.
Response to recommendation C. Thank you for your valuable suggestion. We have duly noted these limitations and have addressed them in the Materials and Methods section of our manuscript.
Discussion The discussion effectively ties together the study's findings, situating them within the broader context of current research. It highlights the protective role of Lactobacillus spp., the impact of dysbiosis on immune response, and the potential of targeted therapies to improve ART outcomes. To further refine and enhance the quality of this discussion, consider the following points:
- Mechanisms Underpinning Observations: The discussion mentions the protective role of Lactobacillus spp. and the detrimental impact of a dysbiotic microbiota. Expanding on the biological mechanisms, possibly by referring to studies that have elucidated how Lactobacillus spp. Influencing immune regulation within the endometrium would add depth. For instance, discussing the specific immune pathways modulated by Lactobacillus spp. and how dysbiosis disrupts these pathways could provide insights into potential therapeutic targets.
Response to recommendation A. Thank you for the apparent oversight in the discussion
‘’Lactic acid produced by Lactobacillus spp. has been shown to elicit an anti-inflammatory response and inhibit the production of proinflammatory cytokines and chemokines triggered by toll-like receptor (TLR) activation in cervical and vaginal epithelial cells under acidic conditions [67,68]. Furthermore, lactic acid can stimulate the secretion of the anti-inflammatory cytokine interleukin (IL)-10, diminish the production of the proinflammatory cytokine IL-12 in dendritic cells, and attenuate the cytotoxicity of natural killer cells [69]. The anti-inflammatory properties of lactic acid are also contingent upon the presence of organic acids synthesized by microorganisms to sustain vaginal health. This is primarily achieved through the upregulation of the anti-inflammatory cytokine IL-1RA, inhibition of the proinflammatory signaling of IL-1 cytokines, and modest reduction in the production of proinflammatory cytokines such as IL-6 and macrophage inflammatory protein 3 alpha [68]. Hence, the interplay among flora, metabolites, and immunity within the healthy reproductive tract is crucial for preserving its overall health. Any imbalance among these components can disrupt the equilibrium of the reproductive tract.’’
- Therapeutic Interventions: The discussion benefits from detailing the outcomes of therapeutic interventions. Further elaboration on why certain therapies were chosen based on the microbiota profile and how these interventions align with current best practices in managing RIF and RPL could enhance understanding. Additionally, discussing the potential for personalized therapeutic approaches based on endometrial microbiota profiling could highlight the study's clinical implications.
Response to recommendation B. Thank you for this recommendation. These information has been added to manuscript.
We have expanded the discussion section to include detailed therapeutic interventions, shedding light on the rationale behind selecting specific therapies based on the microbiota profile. Furthermore, we have highlighted the potential for personalized therapeutic approaches based on endometrial microbiota profiling. This method of studying dysbiotic microorganisms allows for a tailored selection of therapy, which holds significant clinical implications for improving treatment outcomes in patients with recurrent implantation failure and recurrent pregnancy loss.
- Addressing the Limitations: The limitations section is crucial for understanding the study's scope and the interpretation of its findings. Expanding on how these limitations might affect the study's conclusions and suggesting specific future research directions to address these gaps would be beneficial. For instance, discussing how the inclusion of a control group of women without reproductive issues could help delineate the normal variations in endometrial microbiota and immune cell dynamics across the menstrual cycle.
Response to recommendation C Thank you for your comment. We have added a section to outline the scope of the study, as recommended.
This study compares endometrial immune cells in women experiencing reproductive issues with different microbiota types. It is important to note that a control group consisting of women with a normal microbiome and without reproductive problems was not included, which limits our ability to fully understand immune cell dynamics across diverse contexts. In addition, including a control group of healthy women of childbearing age without reproductive issues could provide valuable insights for selecting appropriate immunomodulatory therapy. Another limitation of our study is that it only focuses on the relative values of decidual immune cells, without considering their functional characteristics and activation status. Furthermore, the lack of information on metabolites is another limitation, as these molecules are essential in evaluating the endometrial microenvironment.
- Implications for Clinical Practice: While the therapeutic outcomes are promising, discussing the practical implications for routine clinical practice would be valuable. This could include considerations for integrating endometrial microbiota analysis in ART protocols, potential challenges in adopting these diagnostic and therapeutic approaches, and standardized guidelines based on microbiota profiles.
Response to recommendation D. Thanks for this comment. Your recommendation has been followed as we have described possible applications in clinical practice in additional point.
Implication for Clinical Practice
Understanding the implications of our findings for clinical practice is paramount. Integrating the analysis of endometrial microbiota into routine clinical assessments can offer valuable insights into optimizing treatment strategies, particularly in assisted reproductive technology (ART) protocols. By leveraging microbiota profiles, clinicians may personalize therapeutic approaches, potentially improving treatment outcomes for patients experiencing reproductive challenges. However, the implementation of microbiota analysis in clinical practice may pose challenges, including the development of standardized guidelines and addressing logistical considerations. Nonetheless, by embracing these insights, healthcare providers can enhance their ability to tailor interventions and improve patient care in reproductive medicine.
February 18, 2024

Reviewer 2 Report
Comments and Suggestions for Authors
The study conducted by Blazheva et al. aims to determine the microbiome patterns in patients with RIF and RPL. Some suggestions should be considered:
- - The introduction should be improved with the different points of view that currently exist in the study of the endometrial microbiome. Different studies using different technologies yield different results. Introducing this issue and the studies (Sola-Leyva 2021 (10.1093/humrep/deaa372) and Winters 2019 (10.1038/s41598-019-46173-0)) should be cited as another perspective on the dominance of lactobacilli.
- - The introduction should be shortened.
- - During sample collection, did the authors flush into the uterine cavity to promote mucosal retrieval?
- - Report the characteristics of the menstrual cycle of the patients in case the cycle phase was not determined using ovulation strips.
- - In section 5 of the methodology, lines 230-250 move to the results.
- - Why do the authors not more concretely evaluate the presence of endometritis using additional markers such as CD8, MUM1? More evidence must be add for CE.
- Report the independent analyses of the endometrial microbiome results for RIF and RPL patients.
- - Authors said that in 2.1, section 7, that microbiological examination of a vaginal swab and cervical canal was performed. Please add this information for all the patients.
The study is well written and articulated; however, some sentences are too long and complex and could be simplified.
Reviewer 3 Report
Comments and Suggestions for Authors
The reviewer believes that this manuscript has shortcomings, without correction of which the manuscript cannot be accepted for publication.
Firstly, the reviewer's comment concerns the study design. The study involved two groups of individuals (with recurrent implantation failure and recurrent miscarriage); The reviewer thinks that the data for these could be presented separately.
The “Introduction” section is cumbersome and does not make it clear why the authors did this research; Moreover, there are such studies. Perhaps this section should have clarified how the authors’ research differs from similar ones.
Lines 120-124. The purpose of the research described in this manuscript may need to be restated to make it easier for readers to understand.
Section "Materials and methods"
Lines 192-198. This fragment should indicate how the authors isolated DNA.
Lines 226-229. This fragment looks like an advertisement for Femoflor®. The reviewer believes that in this case it is sufficient to indicate the manufacturer of the Femoflor® reagent kit; in addition, it seems that the primers that are used in this set of reagents should be described or a link to such a description should be provided.
Section "Results"
Line 300-303. Here we are talking only about the data presented in table 1, so the reference to table 2 should be removed.
The reviewer believes that the data in Table 1 may be misleading to readers; here it is only clear that in the fourth group the number of anaerobes and Candida spp. has increased (apparently significantly, but statistics are not shown in the table). In the “materials and methods” section it is indicated that the set of reagents used by the authors allows (lines 223-229 ) determine not only the amount of total bacterial mass, but also the number of microorganisms such as “... Lactobacillus spp., Enterobacteriaceae, Streptococcus spp., Staphylococcus spp., Gardnerella vaginalis / Prevotella bivia / Porphyromonas spp., Eubacterium spp., Sneathia spp. /Leptotrichia spp. /Fusobacterium spp., Megasphaera spp./Veilonella spp. /Dialister spp., Lachnobacterium spp. /Clostridium spp., Mobiluncus spp. /Corynebacterium spp., Peptostreptococcus spp., Atopobium vaginae, Mycoplasma hominis, Ureaplasma (urealyticum and parvum), Candida spp. and presence of Mycoplasma genitalium". Table 1 lists only aerobic and anaerobic bacteria, Candida spp. and combinations of these groups of microorganisms. Why aerobes or anaerobes, why not, for example, microaerophiles? How do you know if species diversity has increased in the fourth group compared to the third or second? What group do lactobacilli belong to? Clearly, Table 1 requires revision to make it easier for future readers to understand what is being discussed in this portion of the manuscript.
Does Figure 1 contain the same data as Table 2? If this is the case, then it seems that authors should choose one way to present their data so that there is no duplication.
In the “materials and methods” section it is indicated (lines 242-244) that patients in groups 2, 3, and 4 were subjected to additional exposure (through the use of antibiotics, antimycotics, probiotics), and that after such exposure, the characteristics of samples from the uterus were examined. Why don't the authors provide this data? Did exposure through treatment with antibiotics, antimycotics, or probiotics lead to any changes in the endometrium? Have patients of groups 2, 3, and 4 become similar to patients of the first group?
In the “materials and methods” section (lines 242-244), the authors write that assisted reproductive technology was applied only to those individuals who did not have dysbiotic microorganisms in the endometrium. Did dysbiotic microorganisms disappear after the first course of treatment for patients of the second, third, and fourth groups? If so, how do the authors explain the striking difference in outcomes with assisted reproductive technology?
Reviewer 4 Report
Comments and Suggestions for Authors
This study is an interesting attempt to determine the relationship between the microbiota of the uterus and the local immune response (maybe not the modern one itself). Currently, there is evidence that not only infectious agents, but also endometrial destruction products are able to support the immune response and chronization of the inflammatory process in the endometrium, which negatively affects the implantation process.
In addition, the endometrium is a biotope with a low biological mass (the estimate of the bacterial load of the uterus is 100-10, 000 times lower than in the vagina), which currently presents difficulties even for studying the uterine microbiome with the use of high-molecular diagnostic methods of bacterial DNA). However, it is still of scientific interest to study the microbiome using sequencing methods (targeted sequencing of the 16S RNA site, complete metagenomic analysis, metatrancriptomic analysis). In this regard, it is possible to invite the authors to discuss in the discussion the advantage of the chosen research method (PCR analysis - femoflor).
Transcervical removal of material from the uterine cavity, although it is the only clinical method for assessing the condition of the endometrium, nevertheless, is inevitably associated with contamination by microorganisms of the cervix and vagina, which is confirmed by a number of studies in which endometrial biopsies were obtained after hysterectomy (Winters et al., 2019), laparoscopy (Chen et al., 2017) and/or during cesarean section (Leoni et al., 2019;).
The researchers concluded that lactobacilli do not dominate the uterine cavity, and bacteria such as Pseudomonas, Acinetobacter, Vagococcus and Sphinogobium make up a large part of the endometrial microbiome. Moreover, 40% of endometrial samples collected after abdominal hysterectomy did not contain such a concentration of microorganisms that would exceed the level of negative control (Winters et al., 2019). Such results reinforce the ongoing debate about the presence of a unique uterine microbiome. (it may be suggested to the authors to include them in the discussion).
Despite the fact that the authors indicated the absence of a control group in the disadvantages (limitations) of the study, this fact, of course, reduces the quality of the study. In addition the presence of plasma cells should be taken into account as the gold standard for chronic endometritis histological diagnosis.
Round 2
Reviewer 2 Report
Comments and Suggestions for Authors
The manuscript has improved significantly. However, there are still some considerations.
- - Have the authors also explored the possibility of performing 16S rRNA sequencing within the samples? It would be a possibility, and it would add a lot of information to the study.
- - The analysis of the vaginal microbiota still remains unclear to me. Was it conducted at the same time, clarifying the antibiotic treatment? The authors should specify the experimental design more clearly in the methods section.
- - A figure outlining the workflow and design of the study would aid in its understanding.
Comments on the Quality of English Language
The manuscript is overall well-written.
Reviewer 3 Report
Comments and Suggestions for Authors
The authors provided comprehensive answers to the reviewer's questions and comments. The reviewer believes that the revision of the manuscript has made it significantly better; the manuscript can be accepted for publication.
There are minor flaws in the text of the manuscript. In lines 336-339 it is unclear what the authors are referring to; in the text of the manuscript instead of a link there is “Error! Reference source not found."
